# ML Reproducibility Challenge 2020
# Learning to Deceive With Attention-Based Explanations

## Reproducibility Summary

**Scope of Reproducibility**

Based on the intuition that attention in neural networks is what the model focuses on, attention is now being used as an explanation for a models' prediction (see Galassi et al. (2020) for a survey). Pruthi et al. (2020) challenge the usage of attention-based explanation through a series of experiments using classification and sequence-to-sequence (seq2seq) models. They examine the model's use of impermissible tokens, which are user-defined tokens that can introduce bias e.g. gendered pronouns. Across multiple datasets, the authors show that with the impermissible tokens removed the model accuracy drops, implying their usage in prediction. And then by penalising attention paid to the impermissible tokens but keeping them in, they train models that retain full accuracy hence must be using the impermissible tokens, but that does not show attention being paid to the impermissible tokens. As the paper's claims have such significant implications for the use of attention-based explanations, we seek to reproduce their results.

**Methodology**

Using the authors' code, for classifiers we attempt to reproduce their embedding, BiLSTM, and BERT results across the occupation prediction, gender identify, and SST + wiki datasets. Further, we reimplemented BERT using HuggingFace's transformer library (Wolf et al., 2019) with restricted self-attention (information cannot flow between permissible and impermissible tokens). For seq2seq we used the authors' code to reproduce results across bigram flip, sequence copy, sequence reverse, and English-German (En-De) machine translation datasets. We performed refactoring on the authors' code aiming toward a more uniformly usable code style as well as porting across to PyTorch Lightning. All experiments were run in approximately 130 GPU hours on a computing cluster with nodes containing Titan RTX GPUs.

**Results**

We reproduced the authors' results across all models and all available datasets, confirming their findings that attention-based explanations can be manipulated and that models can learn to deceive. We also replicated their BERT results using our reimplemented model. There was only one result not as strongly ($> 1$ S.D.) in their experimental direction.

**What Was Easy**

The authors' methods were largely well described and easy to follow, and we could quickly produce the first results as their code worked straightaway with minor adjustments. They were also extremely responsive and helpful via email.

**What Was Difficult**

Re-implementing the BERT-based classification model to perform replicability, with further specification details on model architecture, penalty mechanism, and training procedure needed. Also, porting code across to PyTorch Lightning.

**Communication With Original Authors**

There was a continuous email chain with the authors for several weeks during the reproducibility work. They made additional code and datasets available per our requests, along with providing detailed responses and clarifications to our emailed questions. They encouraged the work and we wish to thank them for their time and support.

# 1 Introduction

Attention is a mechanism to automatically learn the relevance of different elements of the input to a model, rather than relying on manual feature engineering, allowing computational learning to focus on important elements (Galassi et al., 2020). Originally introduced to natural language processing (NLP) for neural machine translation (Bahdanau et al., 2014) its usage has since expanded. Vaswani et al. (2017) termed it "all you need", having removed recurrence and convolutions and relied on attention in their Transformer architecture. Transformers are now in wide use, with Devlin et al. (2018)'s Bidirectional Encoder Representations from Transformers (BERT) a commonly used model in NLP.

Because neural networks are subsymbolic with knowledge stored numerically, it is challenging to understand their inner workings (Galassi et al., 2020). With interpretability a growing concern in NLP, there is a body of work on attention-based explanations of neural architectures using visualisation of attention weights (Serrano and Smith, 2019). However, there is a rich and ongoing debate about whether attention is an explanation or not (Jain and Wallace, 2019; Wiegreffe and Pinter, 2019). Acknowledging the debate, Pruthi et al. (2020) whose work we seek to reproduce, examine whether models can learn to deceive, by adding a penalty to the loss function that punishes the model when attention is paid to impermissible tokens. These tokens are user-defined and may refer inter alia to terms for protected traits such as gender (the pronouns *she*, *her* etc.), sexual orientation, or race. Their research indicates that the impermissible tokens are still being used by the model as there is no accuracy drop seen, while there is one when these tokens are instead fully removed. Thus, the model is both able to use the impermissible tokens in learning and inference, but not pay attention to them. Hence bringing into question the validity of using attention in the explanation of a model's decision.

# 2 Scope of Reproducibility

The core finding of the paper is that attention-based explanations of models can be deceptive by, for instance, hiding the model's use of gendered pronouns at inference from an auditor. Specifically, the authors show that attention weights can be manipulated during training by penalising the allocation of attention to impermissible tokens, without this affecting model performance. Any resulting attention-based explanation might suggest that the model did not rely on impermissible tokens to make its predictions, when in reality the model still uses these but not through the attention mechanism, thereby making the model "deceptive". The key findings can be decomposed into the following claims which we are testing in our research:

1. The attention mass on impermissible tokens can be reduced without significantly affecting the classification accuracy of Embeddings + Attention, BiLSTM + Attention, and BERT + Attention models across several tasks (see Table 3).

2. The attention mass on impermissible tokens can be reduced without significantly affecting the seq2seq performance on translation as measured by BLEU (see Table 5).

3. The attention mass on impermissible tokens can be reduced without significantly affecting the seq2seq accuracy on synthetic data tasks (see Table 4).

We chose not to perform the human study, testing whether visualised attention weights could deceive NLP/ML trained and Transformer knowledgeable participants, as we deemed that the small sample size does not add value to the results.

# 3 Methodology

Initially, we attempted to reproduce the findings from the provided repository without contacting the authors. The authors use three classification models: embeddings with attention; BiLSTM with attention; and a BERT-based model with attention. For the seq2seq tasks, the model is an encoder-decoder architecture. Code for all models (except BERT) was available in the authors' repository, and aside from minor dependency issues in the environment file, we were able to successfully run experiments to reproduce the results. We re-implemented the BERT-based model to replicate their results, and after the BERT-based code was added to the repository, we also reproduced their results. Lastly, we refactored the existing code-base and ported the PyTorch-based code to PyTorch Lightning.

## 3.1 Attention Manipulation

To explicitly optimise the models to learn deceptive attention weights, the authors introduce an auxiliary loss component that penalises the model for attending to impermissible tokens. Impermissible tokens are user-defined from a corpus and are the set of words $\mathcal{I}$ that a model should not use during training or inference, as they might introduce bias or other ethical issues. An example is the use of gendered pronouns "*her, she, Ms.*" which might lead a model to

discriminate against a specific gender. The remaining words in the corpus are deemed permissible, and thus constitute the complement set $\mathcal{I}^c$. Assuming an input sequence $S = w_1, w_2, ..., w_n$ of $n$ tokens, the authors proceed to define a binary attention mask vector $m$ of length $|S|$, with each element denoting the occurrence of an impermissible token:

$$\boldsymbol{m}_i = \begin{cases} 1, & \text{if } w_i \in \mathcal{I} \\ 0 & \text{otherwise} \end{cases}$$

Furthermore, we assume an attention vector $\boldsymbol{\alpha} \in [0, 1]^n$ which denotes the allocated attention for each token in the input sequence. From this, the authors construct the additive task-agnostic penalty term $R$, such that $L' = L + R$, where $R$ captures the extent to which a model's attention layer is penalised for allocating attention to impermissible tokens:

$$R = -\lambda \log(1 - \boldsymbol{\alpha}^T \boldsymbol{m})$$

Here, the $\boldsymbol{\alpha}^T \boldsymbol{m}$ term denotes the attention allocated to impermissible words. Taking the negative log of the complement of this term then allows us to minimise this quantity via standard gradient descent. Furthermore, $\lambda$ is a coefficient that is used to control the extent to which impermissible attention allocation is penalised. Following the authors' methods, we consider values for $\lambda = \{0, 0.1, 1.0\}$. For models featuring multi-head attention (such as BERT), the authors use two different penalty variants. Namely, $R_{mean}$ optimises the mean of the penalty over the set of all heads $\mathcal{H}$, while $R_{max}$ instead only considers the head which allocates the most attention to impermissible tokens:

$$R_{mean} = -\frac{\lambda}{|\mathcal{H}|} \sum_{h \in \mathcal{H}} \log(1 - \boldsymbol{\alpha}_h^T \boldsymbol{m}), \qquad R_{max} = -\lambda \cdot \min_{h \in \mathcal{H}} \log(1 - \boldsymbol{\alpha}_h^T \boldsymbol{m})$$

Note that for the multi-head penalties, only the heads from the model's last layer are considered, thus in BERT $\alpha$ is defined as the attention paid by the [CLS] token to the other tokens.

## 3.2 Model Descriptions

**Embedding + Attention** This model serves as a baseline to the other models used for classification. It contains between 2.7M and 6.5M parameters depending on the dataset and consists of a dot-product attention mechanism applied on word embeddings. The resulting attention vector is then passed into a linear classifier followed by a softmax activation. The size of the embedding in the original paper was 128. We use cross-entropy loss and the Adam optimizer with a learning rate of 0.001 and no weight decay. Pruthi et al. (2020) argue that the accuracy of the Embedding and BiLSTM models could have been greatly impacted by the lambda parameter because those models might be under-parameterised for the SST-Wiki dataset. To study this we also train models with embedding sizes of 256 and 512.

**BiLSTM + Attention** The model still uses word embeddings and consists of a dot-product attention mechanism applied on the output of a bidirectional LSTM (Graves and Schmidhuber, 2005) The resulting attention vector is then passed into a linear classifier followed by a softmax activation. Again, the size of the embedding is 128, but we also train models with embedding sizes of 256 and 512. Its number of parameters is between 2.8M and 6.6M parameters depending on the size of the vocabulary of the given dataset. The BiLSTM was trained using the same hyper-parameters as the Embedding model above, with the dimension of the hidden state being 64.

**Transformer Models** For the transformer-based architecture, we use BERT (Devlin et al., 2018). Specifically, we use a pre-trained instance of `BERT-base-uncased`, which consists of 12 transformer blocks (each with 12 heads) amounting to 109M trainable parameters. We trained each model for 10 epochs, with a batch size of 32. All models were optimised using Adam, with a learning rate of $5e - 5$. Furthermore, we applied dropout with $p = 0.3$ to improve model generalisation. A sequence classification layer is added on top of the architecture, to adapt the model for sentence classification. Following the authors' methods, we apply a self-attention mask $M$ to the self-attention probabilities via element-wise multiplication in the models' forward pass, to avoid information flowing between the sets of impermissible tokens $\mathcal{I}$ and permissible tokens $\mathcal{I}^c$. Specifically, $M$ is a binary matrix of size $n \times n$, where $n$ denotes the sequence length, and where elements $M_{p,q}$ are 1 if both tokens $w_p$ and $w_q$ belong to the same set (either $\mathcal{I}$ or $\mathcal{I}^c$), and 0 otherwise. Moreover, the first column of $M$, which denotes the extent to which all other tokens attend to the [CLS] token, is zero also, as this further restricts the flow of information between tokens from $\mathcal{I}$ and $\mathcal{I}^c$ via the [CLS] token.

**Seq2seq** Pruthi et al. (2020) provide a bidirectional and unidirectional Gated Recurrent Unit (GRU) with dot-product attention respectively for their encoder-decoder model tackling seq2seq tasks. The input is passed through the encoder and decoder, where the final hidden state from the bidirectional GRU fed through a linear layer is the initial hidden state to the decoder. The embedding size was 256 and hidden size 512 for both encoder and decoder. We also use a teacher forcing ratio of 0.5 as well as the top-1 greedy strategy for decoding output sequences. For baseline experiments, this model is also trained with no attention and uniform attention overall source tokens. It overall contains 8.7M parameters for the synthetic datasets and 48.55M parameters for the Multi30K dataset, which are both described in Section 3.3.

Table 1: Details of datasets.

| Task Type | Dataset | Examples | Train | Val | Test | Label Dist. |
|---|---|---|---|---|---|---|
| Classification | Occupation Prediction | 25185 | 17629 | 2519 | 5037 | 68-32 |
| | Gender Identity | 11271 | 9017 | 1127 | 1127 | 50-50 |
| | SST + Wiki | 9613 | 6920 | 872 | 1821 | 48-52 |
| Seq-to-Seq | Bigram Flip | 300000 | 100000 | 100000 | 100000 | - |
| | Sequence Copy | 300000 | 100000 | 100000 | 100000 | - |
| | Sequence Reverse | 300000 | 100000 | 100000 | 100000 | - |
| | En-De Translation | 31016 | 29001 | 1015 | 1000 | - |

## 3.3 Datasets

The original work features 8 tasks with associated datasets. For the classification models, these are Occupational Prediction, Gender Identity, SST + Wiki, and Reference Letters. For the seq2seq experiments, three synthetic datasets were used; Bigram Flip, Sequence Copy, and Sequence Reverse tasks. Additionally, the Multi30K dataset (Elliott et al., 2016) was used for English to German machine translation (MT). All of the datasets were available in the authors' repository except for Reference Letters, with the authors citing privacy concerns. Consequently, we were not able to reproduce this experiment. For Occupation Prediction the authors state that downsampling by a factor of 10 was done for minority classes. As it was not clear from the data provided in the repository if downsampling had already been applied, the authors confirmed via email that this was the case. No further pre-processing was required, besides that already present in the authors' code. Details of the datasets used in our experiments are provided in Table 1.

## 3.4 Hyperparameters

Except for the $\lambda$ coefficient (values 0.0, 0.1, and 1.0) as used in the computation of the regularising component $R$, the original work did not provide details regarding hyperparameters and/or tuning thereof. Upon contacting the authors, we learned that no hyperparameter tuning was performed, as the experimental findings could be achieved with conventional parameters. Therefore, in reproducing their experiments we have used the same standard configurations as the authors.

## 3.5 Experimental Setup and Code

The code used to reproduce the experiments can be found in this Github repository [1]. The authors' negative baseline, the first row of each model in Table 3 of the original paper, was produced by removing the impermissible tokens (anonymising or deleting). They show that the performance of the model dropped. This drop was in comparison to the true baseline in row two, which provides the models' performance when impermissible words are freely used with no manipulation penalty applied i.e. $\lambda = 0.0$. The third and fourth rows provide results for adjusting the penalty coefficient to 0.1 and 1.0 respectively. To reproduce the experiments by anonymising or removing the impermissible tokens, we had to look deeper into their script. For the Occupation Prediction and Gender Identity datasets, the authors provided an anonymisation functionality that transformed all pronouns to gender-neutral ones. However, for the SST + Wiki dataset, we had to implement the functionality to remove the SST sentence because it was not present in the scripts. Similarly, we added the functionality to the training script for the provided BERT implementation.

The repository contained a bash script to run the experiments with Embedding + Attention and BiLSTM + Attention without removing or anonymising the impermissible tokens. We recreated the classification experiments using the seed values from this script. The training outputs of those experiments were not as presented in the README of the authors' repository and did not contain a clear attention mass value. However, the authors clarified that the "attention ratio" measure was used in the paper. From those experimental runs, we could determine the average and standard deviation of the five runs.

For the classification models, we chose to largely follow the authors' experimental setup: we run experiments with the same set of values for the loss coefficient $\lambda \in 0, 0.1, 1.0$. Furthermore, all models are trained for 10 epochs and are evaluated on the development set after each epoch. Here, we measure two metrics; the validation accuracy, and the average attention mass over all examples. The model with $\lambda = 0$ serves as the baseline for the 'adversarial' models with $\lambda = 0.1, 1.0$, i.e. the models that are explicitly optimised to learn deceptive attention maps. For the BERT replication, when evaluating models on the test set, we follow the authors' heuristic, i.e. we select the checkpoint which is within 2% of the baseline test accuracy, and which has the greatest reduction in attention mass on the validation set.

---

[1] https://github.com/MatPrst/FACT

Table 2: Breakdown of approximate computational requirements for running experiments per task, for a single seed.

| Classification | | GPU Hours | | | |
|---|---|---|---|---|---|
| Model | Batch size | Occupation Pred. | Gender Identity | SST + Wiki | |
| Embedding | 1 | 0.62 | 0.56 | 1.1 | |
| BiLSTM | 1 | 0.94 | 1 | 1.3 | |
| BERT | 32 | 3.1 | 1.5 | 1.2 | |
| BERT(HF) | 32 | 4.8 | 1.9 | 2.5 | |
| Seq-to-Seq | | GPU Hours | | | |
| Model | Batch size | Bigram Flip | Sequence Copy | Sequence Reverse | En-De |
| Enc-Dec | 128 | 0.42 | 0.36 | 0.34 | 0.15 |

As per the authors' experimental setup we considered the four sequence-to-sequence tasks: Bigram Flipping, Sequence Copying, and Sequence Reversal are synthetic tasks that work with input-output-mappings with the respective gold alignments considered as impermissible tokens. The models are trained on 100K random input sequences with length 32 from a vocabulary of 1000 tokens and validated and tested on 100K unseen random sequences. Machine translation from German to English acts as the fourth task for which gold alignments are not available. Thus, the Fast Align toolkit (Dyer et al., 2013) was used by the authors to align target and source words. In this task, the aligned words are used as impermissible tokens.

The seq2seq experimental results in Table 4 of the original paper provide the averaged value over five runs. The different runs and their results were not available in the repository, however, after emailing the authors we were provided with the results for each of the five experimental runs in each condition, along with the seeds used. This allowed us to recreate the experiment using the same seeds, and to determine along with the average, whether the standard deviation between our results also matched.

Pertaining to the English to German translations, the BLEU score was used, however, this was not available in the repository. After contacting the authors, we were provided with a link to the BLEU library they had used which is called "compare-mt" (Neubig et al., 2019). We had meanwhile used the NLTK implementation (Bird et al., 2009), presuming it to be the most likely used. Therefore, for translation, we have used two different BLEU implementations: compare-mt for reproduction and NLTK for replication.

### 3.6 Computational Requirements

All experiments were run on the LISA computing cluster provided by SURFsara, which is available to University of Amsterdam Master students. The nodes used contained 4 x Titan RTX GPUs. A breakdown of the computation is provided in Table 2.

## 4 Reproduction and Replication Results

### 4.1 Classification

For the classification results, see Table 3. As can be observed, our results support claim 1, and we were able to reproduce the results from Table 3 in the original paper (excluding the Reference Letters dataset). Furthermore, we replicated the BERT(max) and BERT(mean) results using our implementation. While the results for the BERT replication match the authors' findings closely, there are also some noticeable differences; particularly, for the SST+WIKI task, we can see that for both the *mean* and *max* models for $\lambda = 1.0$, the replication of BERT does not manage to retain its performance, with a drop in accuracy of 4 and 6 percent, respectively.

### 4.2 Seq2seq

The results in Table 4 and Table 5 support claims 2-3, and we were able to reproduce the results from Table 4 in the original paper. Especially the reported mean accuracy by Pruthi et al. (2020) shows no significant difference to our reported values for all seq2seq tasks except for the baseline experiments with uniform and no attention for the tasks sequence copy and sequence reverse. Both have a mean difference of 3-14 % accuracy regarding the authors' accuracies. Pruthi et al. (2020) did not report accuracies for the translation task in their original paper, but they provided us with

Table 3: Classification results from Table 3 in Pruthi et al. (2020) with cell scheme *author | reproduced* for all models except BERT(HgFc) which follows cell scheme *author | replicated*. Our values are means over 5 different seeds.

| Model | $\lambda$ | I | Occupation Pred. | | Gender Identity | | SST + Wiki | |
|---|---|---|---|---|---|---|---|---|
| | | | Acc. | A.M. | Acc. | A.M. | Acc. | A.M. |
| Embedding | 0.0 | X | 93.8 \| 93.4 | - | 66.8 \| 71.0 | - | 48.9 \| 49.3 | - |
| Embedding | 0.0 | ✓ | 96.3 \| 96.5 | 51.4 \| 56.4 | 100 \| 100 | 99.2 \| 90.1 | 70.7 \| 68.1 | 48.4 \| 49.9 |
| Embedding | 0.1 | ✓ | 96.2 \| 96.3 | 4.6 \| 5.70 | 99.4 \| 99.9 | 3.4 \| 8.8 | 67.9 \| 69.5 | 36.4 \| 16.9 |
| Embedding | 1.0 | ✓ | 96.2 \| 96.1 | 1.3 \| 1.50 | 99.2 \| 99.5 | 0.8 \| 4.6 | 48.4 \| 51.8 | 8.70 \| 12.9 |
| BiLSTM | 0.0 | X | 93.3 \| 93.6 | - | 63.3 \| 71.1 | - | 49.1 \| 48.9 | - |
| BiLSTM | 0.0 | ✓ | 96.4 \| 96.7 | 50.3 \| 44.1 | 100 \| 100 | 96.8 \| 95.5 | 76.9 \| 75.9 | 77.7 \| 81.5 |
| BiLSTM | 0.1 | ✓ | 96.4 \| 96.6 | 0.08 \| 3.70 | 100 \| 100 | $< 10^{-6}$ \| 0.07 | 60.6 \| 65.1 | 0.04 \| 0.99 |
| BiLSTM | 1.0 | ✓ | 96.7 \| 96.5 | $< 10^{-2}$ \| 0.015 | 100 \| 100 | $< 10^{-6}$ \| 0.0047 | 61.0 \| 64.9 | 0.07 \| 0.035 |
| BERT | 0.0 | X | 95.0 \| 95.8 | - | 72.8 \| 82.3 | - | 50.4 \| 50.2 | - |
| BERT(mean) | 0.0 | ✓ | 97.2 \| 97.1 | 13.9 \| 9.10 | 100 \| 99.9 | 80.8 \| 55.1 | 90.8 \| 91.8 | 59.0 \| 17.4 |
| BERT(mean) | 0.1 | ✓ | 97.2 \| 97.3 | 0.001 \| 0.007 | 99.9 \| 99.9 | $< 10^{-3}$ \| 0.004 | 90.9 \| 91.5 | $< 10^{-2}$ \| 0.04 |
| BERT(mean) | 1.0 | ✓ | 97.2 \| 97.3 | $< 10^{-3}$ \| 0.0007 | 99.9 \| 99.9 | $< 10^{-3}$ \| 0.0003 | 90.6 \| 91.9 | $< 10^{-3}$ \| 0.005 |
| BERT | 0.0 | X | 95.0 \| 95.8 | - | 72.8 \| 82.3 | - | 50.4 \| 50.2 | - |
| BERT(max) | 0.0 | ✓ | 97.2 \| 97.1 | 99.7 \| 65.5 | 100 \| 99.9 | 99.7 \| 99.8 | 90.8 \| 91.8 | 96.2 \| 67.4 |
| BERT(max) | 0.1 | ✓ | 97.1 \| 97.1 | $< 10^{-3}$ \| 0.007 | 99.9 \| 99.9 | $< 10^{-3}$ \| 0.003 | 90.7 \| 91.9 | $< 10^{-2}$ \| 0.04 |
| BERT(max) | 1.0 | ✓ | 97.4 \| 97.2 | $< 10^{-3}$ \| 0.0008 | 99.8 \| 99.9 | $< 10^{-4}$ \| 0.0005 | 90.2 \| 91.8 | $< 10^{-3}$ \| 0.003 |
| BERT(HgFc) | 0.0 | X | 95.0 \| 95.2 | - | 72.8 \| 81.2 | - | 50.4 \| 52.8 | - |
| BERT(mean) | 0.0 | ✓ | 97.2 \| 97.1 | 13.9 \| 23.16 | 100 \| 99.9 | 80.8 \| 57.6 | 90.8 \| 91.2 | 59.0 \| 68.66 |
| BERT(mean) | 0.1 | ✓ | 97.2 \| 96.8 | 0.001 \| 0.006 | 99.9 \| 99.9 | $< 10^{-3}$ \| 0.001 | 90.9 \| 90.7 | $< 10^{-2}$ \| 0.018 |
| BERT(mean) | 1.0 | ✓ | 97.2 \| 97.1 | $< 10^{-3}$ \| 0.002 | 99.9 \| 99.8 | $< 10^{-3}$ \| 0.001 | 90.6 \| 85.4 | $< 10^{-3}$ \| 0.019 |
| BERT(HgFc) | 0.0 | X | 95.0 \| 95.2 | - | 72.8 \| 81.24 | - | 50.4 \| 52.8 | - |
| BERT(max) | 0.0 | ✓ | 97.2 \| 97.2 | 99.7 \| 66,88 | 100 \| 99.9 | 99.7 \| 93.7 | 90.8 \| 91.1 | 96.2 \| 93.94 |
| BERT(max) | 0.1 | ✓ | 97.1 \| 97.0 | $< 10^{-3}$ \| 0.003 | 99.9 \| 99.8 | $< 10^{-3}$ \| 0.006 | 90.7 \| 89.8 | $< 10^{-2}$ \| 0.007 |
| BERT(max) | 1.0 | ✓ | 97.4 \| 97.0 | $< 10^{-3}$ \| 0.001 | 99.8 \| 99.9 | $< 10^{-4}$ \| 0.001 | 90.2 \| 86.0 | $< 10^{-3}$ \| 0.0007 |

Table 4: Seq2seq synthetic data tasks from Table 4 in Pruthi et al. (2020) with cell scheme *author | reproduced*. All values are means over 5 different seeds. Standard deviations are presented in Table 7 of the Appendix.

| Attention | $\lambda$ | Bigram Flip | | Sequence Copy | | Sequence Reverse | |
|---|---|---|---|---|---|---|---|
| | | Acc. | A.M. | Acc. | A.M. | Acc. | A.M. |
| Dot-Product | 0.0 | 100 \| 100 | 94.5 \| 93.9 | 99.9 \| 100 | 98.8 \| 94.1 | 100.0 \| 100 | 94.1 \| 94.0 |
| Uniform | 0.0 | 97.8 \| 95.1 | 5.2 \| 4.71 | 93.8 \| 79.3 | 5.2 \| 4.73 | 88.1 \| 80.8 | 4.7 \| 7.74 |
| None | 0.0 | 96.4 \| 96.4 | - | 84.1 \| 87.3 | - | 84.1 \| 87.2 | - |
| Manipulated | 0.1 | 99.9 \| 100 | 24.4 \| 15.2 | 100.0 \| 100 | 27.3 \| 10.7 | 100 \| 100 | 27.6 \| 16.3 |
| Manipulated | 1.0 | 99.8 \| 99.6 | 0.03 \| 0.01 | 92.9 \| 99.9 | 0.02 \| 0.014 | 99.8 \| 99.9 | 0.01 \| 0.014 |

Table 5: Reproductions (means over 5 different seeds) En-De MT tasks from Table 4 in Pruthi et al. (2020) with cell scheme *author | reproduced*. The BLEU(NLTK) values are not contained in the original paper, thus replicated. Standard deviations are presented in Table 8 of the Appendix.

| Attention | $\lambda$ | BLEU (C-MT) | BLEU (NLTK) | Accuracy | Attention Mass |
|---|---|---|---|---|---|
| Dot-Product | 0.0 | 24.42 \| 24.89 | 24.89 | 36.99 \| 36.75 | 20.66 \| 24.52 |
| Uniform | 0.0 | 18.49 \| 18.37 | 18.37 | 32.31 \| 31.76 | 5.96 \| 5.96 |
| None | 0.0 | 14.89 \| 15.88 | 15.88 | 29.73 \| 30.36 | - |
| Manipulated | 0.1 | 23.69 \| 24.30 | 24.30 | 36.28 \| 36.49 | 7.02 \| 16.77 |
| Manipulated | 1.0 | 20.66 \| 21.07 | 21.07 | 33.82 \| 33.68 | 1.16 \| 1.40 |

additional raw data which also contained the accuracy scores from their experiments. Therefore we also compare accuracies for the En-De MT task.

## 5 Results Beyond Original Paper

The authors state that their seq2seq results in Table 4 of the original paper are based on the average of 5 different seeds. Additional to their work we have examined the standard deviations alongside the average for all the results, comparing their results and ours. Further, while the authors do not state whether classification results from Table 3 in their original paper are based on the average of 5 seeds, we have again completed 5 runs of the experiments and provided the average.

### 5.1 Under-Parameterised Models

In the original paper, the classification results for the Embedding and BiLSTM models for the task on SST+Wiki are outliers because, while the attention mass over the impermissible tokens decreases as $\lambda$ increases, the test accuracy also decreases significantly. The authors speculate that this behavior is due to the models being under-parameterised. We investigated this by training Embedding and BiLSTM models with larger embedding dimensions. In particular, we compared embedding sizes of 128 (original size), 256, and 512. The results are presented in Table 6. Increasing the dimensionality of the embedding does not seem to prevent the accuracy from dropping for larger values of $\lambda$. We speculate that the drop in accuracy is due to the way the impermissible tokens are defined for the SST+Wiki dataset. All the words belonging to the SST sentence are labeled as impermissible and will therefore be penalised by the auxiliary loss component. Because the Wikipedia sentence does not provide useful information for the sentiment prediction, the model cannot rely on it and the accuracy reduces as the penalty term increases. This behaviour does not occur for the other datasets because only a few impermissible tokens were selected for the experiments, allowing the model to find other proxy tokens carrying information about the respective classification tasks. For example, words such as "lesbian" or gendered names such as "Mark" were not labeled as impermissible in the Occupation Prediction dataset. An alternative experiment could have been to define the impermissible tokens of the SST+Wiki dataset as the words in the SST sentence with the strongest positive or negative sentiment scores.

Table 6: Influence of the embedding size for Embedding and BiLSTM models on the SST + Wiki dataset. The values reported are the means over 5 different seeds and the standard deviations.

| | | | Embedding dimension | | | | | |
| Model | $\lambda$ | I | 128 (original size) | | 256 | | 512 | |
| | | | Acc. | A.M. | Acc. | A.M. | Acc. | A.M. |
|---|---|---|---|---|---|---|---|---|
| Embedding | 0.0 | ✓ | $68.1 \pm 1.6$ | $49.9 \pm 2.2$ | $69.1 \pm 2$ | $50 \pm 1.1$ | $68.1 \pm 2.4$ | $51.8 \pm 3.2$ |
| Embedding | 0.1 | ✓ | $69.5 \pm 1.4$ | $17 \pm 1$ | $69.1 \pm 1.6$ | $40.3 \pm 0.48$ | $68.3 \pm 0.76$ | $41.4 \pm 2.3$ |
| Embedding | 1.0 | ✓ | $51.8 \pm 1.1$ | $12.9 \pm 2$ | $51 \pm 0.84$ | $12.3 \pm 1.4$ | $51.8 \pm 2$ | $11.1 \pm 2.2$ |
| BiLSTM | 0.0 | ✓ | $76.4 \pm 0.8$ | $81.5 \pm 7.5$ | $76 \pm 2.7$ | $85.5 \pm 3$ | $76.2 \pm 2.8$ | $76.8 \pm 2.4$ |
| BiLSTM | 0.1 | ✓ | $65.1 \pm 4$ | $0.99 \pm 1.3$ | $65.4 \pm 4.7$ | $0.94 \pm 0.8$ | $62.3 \pm 4.1$ | $3.5 \pm 1.9$ |
| BiLSTM | 1.0 | ✓ | $64.9 \pm 3.1$ | $0.035 \pm 0.02$ | $62.1 \pm 2.5$ | $0.11 \pm 0.09$ | $57.3 \pm 1.5$ | $0.13 \pm 0.9$ |

## 6 Discussion

Our results reproduce Pruthi et al. (2020)'s finding that models can learn to deceive. Jain and Wallace (2019) note that for attention to be an explanation, a different configuration of attention weights for the same piece of text should lead to different predictions. The research which we have reproduced implies that the same accuracy (hence prediction) can be maintained while explicitly changing the configuration of attention weights. The implications are clear; either it is providing further evidence for why attention should not be thought of as an explanation, supporting Serrano and Smith (2019)'s findings that attention weights can be largely zeroed out without affecting accuracy. Or, if attention is an explanation, then models can be still be trained to change the attention-based explanation given and deceive algorithmic audit. This research thus provides a new vein of the investigation into the attention-based explanation debate.

Examination of the standard deviations showed whether reproduction differences were meaningful for seq2seq tasks. Regarding synthetic data, it showed some variance from the authors' values for attention mass, but it was more strongly in the experimental direction, thus supporting their findings. For machine translation, one result at $\lambda = 0.1$ for attention mass was not as strong as their result, but still trending in the experimental direction. While the standard deviation does

show that there is some variance inherent in the reduced attention masses under manipulation, it provides still further robustness to the findings.

Finally, we obtain very similar results with our replication of the authors' BERT-based classification model, for both the *mean* and *max* penalties, for the majority of the tasks. The only results which we did not manage to exactly replicate concern the test accuracies for *SST+Wiki*. However, given that we did manage to *reproduce* these results with the authors' implementation of BERT, it is more likely that the differences in test accuracies between the reproduction and replication experiments can be attributed to slight differences in hyperparameter settings between the two - however, further investigation would be required to confirm this. Further to this, we observe somewhat large differences between the attention masses - particularly for values where $\lambda = 0$. However, given that similar differences persist also for our *reproduction* results of BERT, it may be the case that these quantities are simply more susceptible to stochasticity and/or training dynamics, which would then explain the observed discrepancies with the authors' findings. Either way, these particular differences do not provide grounds to reject the authors' hypotheses; we still observe that for most replication runs, the accuracy is not impacted despite substantial reductions in attention masses on the impermissible tokens. To this end, our replication efforts provide further evidence for the authors' claims that attention-as-explanation can be deceptive.

## 6.1 What Was Easy

Overall, we could easily understand and follow the authors' descriptions of their methods in the paper. An example of this is the way seq2seq tasks and the datasets used were described in a short but comprehensive way. The provided bash scripts revealed the basic setup and could almost immediately be used to run the experiments on our infrastructure described in Section 3.6. Therefore, we were quickly able to reproduce the first results. The codebase for seq2seq tasks was easily restructured into functions, instead of keeping the train and evaluation functionality at a global code level. Besides this, applying PEP8 to the codebase was an easy task - another positive is that restructuring did not break the code massively at any point, which in our opinion testifies to an already consistent architecture. The authors' integration of the BLUE score implementation (compare-mt) was missing. However, we could easily add the NLTK BLEU score implementation into the code. We could further observe robust BLEU score results, like the ones we reproduced and replicated turned out to be not significantly different from those reported by the authors.

## 6.2 What Was Difficult

To a certain extent, it was achievable to re-implement the BERT-based model using only the information provided in the original work. Nevertheless, in the process of re-implementing, several ambiguities arose that we were initially unable to resolve and which had to be clarified by the authors. For instance, the penalty mechanism used to compute $\mathcal{R}$ by default assumed an attention *vector* $\alpha$, while BERT, by default, outputs self-attention *matrices*. We were initially unsure how to obtain a vector from this matrix; only after contacting the authors, we learned that $\alpha$ can be obtained from BERT's self-attention matrices by only considering the first row of the matrix (for a given self-attention head), which represents the extent to which the [CLS] token attends to all other tokens in the sentence. The authors also further clarified that they only used the attention output of the last (12th) transformer block of the model, whereas we initially understood from the paper that all layers should be taken into account.

Additionally, there was some brief confusion relating to the evaluation procedure. In their papers' Section 5.1, the authors provide a heuristic to "pick" the best deceptive model. Based on how this procedure was formulated, we initially believed that multiple models were trained for a given epoch sequentially, after which the best model (as evaluated by admissible accuracy and largest reduction in attention mass) was selected for the next epoch. However, after corresponding with the authors, we learned that rather, a single model is simply trained for 10 epochs, and only then the selection heuristic is applied manually to determine which model *checkpoint* will be used for evaluation on the test set. In these respects, replicating the model architecture, penalty mechanism, and training procedure might have been easier, had the original work been more precise and explicit regarding its methods.

We also chose to port the code to PyTorch Lightning to make it easier to reproduce the research in the future, but this necessitated changes to data loading, pre-processing, and batching. A specific challenge was PyTorch lightning not yet supporting checkpointing over multiple metrics out-of-the-box, meaning we had to implement the authors' multi-metric heuristic ourselves.

## 6.3 Communication With Original Authors

Following our initial contact email, the authors made themselves readily available, quickly responding to a series of emails over multiple weeks, answering all questions clearly, and providing access to everything we required to reproduce their results. We have also provided the authors with this full report.

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

# A Appendix

Table 7: Reproductions seq2seq synthetic data tasks from Table 4 in Pruthi et al. (2020)with cell scheme *author | reproduced*. All values are standard deviations over 5 different seeds.

| Attention | λ | Bigram Flip | | Sequence Copy | | Sequence Reverse | |
|---|---|---|---|---|---|---|---|
| | | Acc. | A.M. | Acc. | A.M. | Acc. | A.M. |
| Dot-Product | 0.0 | 0.00 \| 0.00 | 1.36 \| 0.21 | 0.004 \| 0.00 | 1.06 \| 0.11 | 0.00 \| 0.00 | 0.08 \| 0.13 |
| Uniform | 0.0 | 0.27 \| 1.90 | 0.00 \| 0.00 | 2.52 \| 4.20 | 0.00 \| 0.00 | 3.80 \| 3.40 | 0.00 \| 0.00 |
| None | 0.0 | 0.94 \| 0.95 | - | 2.91 \| 6.40 | - | 6.00 \| 3.40 | - |
| Manipulated | 0.1 | 0.004 \| 0.00 | 22.6 \| 7.50 | 0.00 \| 0.00 | 9.85 \| 9.70 | 0.00 \| 0.00 | 17.7 \| 10.0 |
| Manipulated | 1.0 | 0.12 \| 0.67 | 0.04 \| 0.03 | 10.5 \| 0.005 | 0.04 \| 0.01 | 0.24 \| 0.04 | 0.01 \| 0.01 |

Table 8: Reproductions En-De MT tasks on Multi30K from Table 4 in Pruthi et al. (2020) with cell scheme *author | reproduced*. The BLEU(NLTK) values are replicated and were not provided by the authors. All values are standard deviations over 5 different seeds.

| Attention | λ | BLEU (C-MT) | BLEU (NLTK) | Accuracy | Attention Mass |
|---|---|---|---|---|---|
| Dot-Product | 0.0 | 1.14 \| 0.95 | 0.95 | 0.68 \| 0.72 | 1.15 \| 1.53 |
| Uniform | 0.0 | 0.87 \| 0.76 | 0.76 | 1.05 \| 0.53 | 0.00 \| 0.00 |
| None | 0.0 | 0.68 \| 1.37 | 1.37 | 0.57 \| 1.05 | - |
| Manipulated | 0.1 | 1.01 \| 1.56 | 1.56 | 0.68 \| 0.96 | 1.04 \| 4.20 |
| Manipulated | 1.0 | 2.25 \| 0.89 | 0.89 | 1.80 \| 0.80 | 1.19 \| 0.47 |

