# OpenReview forum: "Learning to Deceive With Attention-Based Explanations"
_ML_Reproducibility_Challenge/2020 — RC2020_

### Official Review · AnonReviewer1 · 2021-03-02
**A good replication study that's clear about strengths and difficulties about reproducing the original work.**

**Rating:** 7
**Confidence:** 4

**Review:**

*Scope of reproducibility:*

The authors clearly state the scope of their experiments (reproducing the embedding experiment, restricted-self attention with BERT and the sequence-to-sequence experiments), and cleanly execute on them.


*Code:*

 In some instances, the authors re-used code from the original authors, and in some instances, they wrote their own. I did not see a reference to the code written for this paper.


*Communication with original authors:*

The paper authors reach out to the original authors. It looks like a good discussion took place, and the original authors were helpful in clarifying some of the ambiguities that arose through the paper (and also in providing code + datasets).


*Hyperparameter Search:*

Neither the replication study nor the original paper used a hyperparameter search. However, the replication study included results on the variance between the 5 random seeds (original paper reported the mean).

*Ablation Study:*

I don't believe the replication study performed any ablations.

*Discussion on results:*

The replication study presented an excellent description of the reproducibility of the original paper and made clear when the results reproduced and did not reproduce. They clearly stated that some details were ambiguous, but that they were ultimately able to resolve those details.

*Recommendations for reproducibility:*

The replication study authors and original paper authors seem to have clarified some ambiguities during their discussion. Those would be useful to add to the original paper.

*Results beyond the paper:*

The replication study investigates the idea that under-parameterization of the models could lead to a decrease in test accuracy when the hyperparameter controlling attention mass on impermissible tokens increases (less able to pay attention to impermissible tokens). They investigated this by increasing the embedding dimension and found it did not improve test accuracy.

*Overall organization and clarity*

* I appreciated the reproduction of the tables from the paper with the author|reproduced! It made it easy to follow along with.
* Thank you for including the breakdown of the computational requirements for running each task (table 2). This is great.
* I think the explanation of the seq2seq tasks makes sense if you have read the original paper, but could be confusing if someone has not. Please try writing more on this!




**Familiar With The Original Paper:**

I have read the original paper

**Reproducibility Summary:**

Report has summary

---

### Official Review · AnonReviewer2 · 2021-03-05
**The author reproduces the result of original paper very precisely. They follow the structure well. there are no result beyond the original paper.**

**Rating:** 8
**Confidence:** 5

**Review:**

The authors follow the Reproducibility Summary very well.
The authors re-used the original code repository. They also ported the code to PyTorch Lightning to make it easier to reproduce the research in the future.
They did not change any hyperparameters and used the same value in the original paper.
The authors had fair contact with the authors of original papers and they had discussed minor issues.
There are no changes in hyperparameters. The implementation results are the average of five-run times of training the model. They used all datasets except the Reference Letters, due to privacy concerns.
The authors added two Blue scores in the seq-seq model for translation.
They do not propose any beyond results or improving the original paper, but they have a good discussion that can describe the deep understanding of the paper.
There are minor grammar typos.

**Familiar With The Original Paper:**

I have read the original paper

**Reproducibility Summary:**

Report has summary

---

### Official Review · AnonReviewer3 · 2021-03-10
**Well written and concise submission**

**Rating:** 8
**Confidence:** 4

**Review:**

The paper is well written and easy to follow. Authors seem to understand the original paper very well and have done a good job in organizing the content. Given one of the classification dataset was not publicly available, reproducing for one task was not feasible. Apart from which authors have experimentally verified the claims on other tasks/datasets.

Authors have been in constant touch with original paper authors and it is appreciable that original authors have helped in reproducing the experiments by providing data/code as requested by authors along with providing more details/clarifying queries.

It is good that the authors had re-implemented BERT model (as the source code was not available initially) and discussed the challenges in re-implementing with information provided in paper which is important for Reproducibility challenge. However, they were not able to replicate the results. Once the source code for BERT experiments was provided by original authors they were able to reproduce the results and verify authors claims. It would have been interesting if the authors could have identified the reason for performance drop in their re-implementation setting.

In table 4, it is interesting to note that there is no change in accuracy for 'Embedding model' on occupation prediction task which is not expected as removing impermissible tokens negatively impacts performance. Any explanation for this behavior w.r.t impermissible tokens ?
Table 5 includes accuracies from paper and reproduced results, however the original paper does not report accuracy for the task 'En-De MT' - where do these numbers come from ?

Additionally, authors experiment with different embedding sizes hoping to counter the performance drop as λ increases - however it was inconclusive. Having said that authors provide another insight into why they believe could be the reason for performance drop. It would have been interesting to see the impact of the way to remove impermissible tokens on performance trend.

Overall, the paper provides good discussion points and clearly outlines what was provided, what was challenging to infer based on the information in paper/code. They perform all the experiments mentioned in the original paper and verify the claims along with providing insights and implementation aspect. This work helps in understanding the internal details required to reproduce the original paper. Hence, I would recommend to accept this paper.

**Familiar With The Original Paper:**

I have read the original paper

**Reproducibility Summary:**

Report has summary

---

### Decision · Program_Chairs · 2021-03-31

**Decision:**

Accept

**Comment:**

Selected for ReScience-C Journal Publication.

The authors of this reproduction reimplemented the original paper in pytorch lightning, which we expect will be valuable to the community. They successfully reproduced the majority of the claims in the original paper, and were clear about the one result that they couldn't reproduce -- that's a particularly useful data point for future researchers who are interested in building upon this work.